# Zingiber Officinale Roscoe: The Antiarthritic Potential of a Popular Spice—Preclinical and Clinical Evidence

**DOI:** 10.3390/nu16050741

**Published:** 2024-03-05

**Authors:** Joanna Szymczak, Bogna Grygiel-Górniak, Judyta Cielecka-Piontek

**Affiliations:** 1Department of Pharmacognosy and Biomaterials, Poznan University of Medical Sciences, Rokietnicka 3, 60-806 Poznan, Poland; jszymczak@ump.edu.pl; 2Department of Rheumatology, Rehabilitation and Internal Diseases, Poznan University of Medical Sciences, 61-701 Poznan, Poland; bgrygiel@ump.edu.pl; 3Department of Pharmacology and Phytochemistry, Institute of Natural Fibres and Medicinal Plants, 60-630 Poznan, Poland

**Keywords:** ginger, osteoarthritis, rheumatoid arthritis, clinical trials

## Abstract

The health benefits of ginger rhizomes (*Zingiber officinale Roscoe*) have been known for centuries. Recently, ginger root has gained more attention due to its anti-inflammatory and analgesic activities. Many of the bioactive components of ginger may have therapeutic benefits in treating inflammatory arthritis. Their properties seem especially helpful in treating diseases linked to persistent inflammation and pain, symptoms present in the course of the most prevalent rheumatic diseases, such as osteoarthritis (OA) and rheumatoid arthritis (RA). This review analyzes the current knowledge regarding ginger’s beneficial anti-inflammatory effect in both in vitro and in vivo studies as well as clinical trials. The drug delivery systems to improve ginger’s bioavailability and medicinal properties are discussed. Understanding ginger’s beneficial aspects may initiate further studies on improving its bioavailability and therapeutic efficacy and achieving more a comprehensive application in medicine.

## 1. Introduction

### 1.1. Inflammatory Arthritis Prevalence and Treatment

Inflammatory arthritis, a group of joint disorders, is a major cause of musculoskeletal pain and decreased mobility. There are numerous subtypes of joint inflammations, among which osteoarthritis (OA) and rheumatoid arthritis (RA) are the most common [1,2]. Approximately 240 million people globally suffer from OA symptoms, including 10% of males and 18% of women over the age of 60 [3], while the estimated global prevalence of RA reached 17.6 million people per year in 2020. By 2050, this number is expected to reach 31.7 million [4]. Even though each type of arthritis has a unique pathomechanism, they all cause a diminished quality of life as a result of pain, which leads to impairment, disability, and early death.

Finding an effective treatment for inflammatory arthritis is of utmost importance in treating patients with OA and RA. Thus, according to the recommendations of the European Alliance of Associations for Rheumatology (EULAR) and the American College of Rheumatology, the preferred treatment option is the use of drugs revealing anti-inflammatory properties, which reduce joint edema and alleviate pain. Unfortunately, this therapy is not without side effects [5]. Non-steroidal anti-inflammatory drugs (NSAIDs), glucocorticoids, and, in the case of RA, disease-modifying antirheumatic drugs (DMARDs) are the most often prescribed medications. Synthetic, conventional DMARDs such as methotrexate, leflunomide, sulfasalazine, and hydroxychloroquine are all first-line treatments. If two conventional DMARDs are ineffective, other medications are used, such as synthetic targeted (Janus kinase [JAK]-inhibitors) or biological DMARDs [anti-cytokine drugs—e.g., tumor necrosis factor inhibitors—anti-TNFα, interleukin-6-inhibitors, and non-anti-cytokine medications such as abatacept or rituximab) [6].

Alternatives to synthetic medications are still being pursued due to the high cost and various adverse effects of DMARDs [7]. Since natural biomolecules have fewer side effects than synthetic medications, their use is increasing in treating non-severe inflammation mainly caused by viral infection or trauma. Moreover, phytopharmaceuticals can be used as an additive therapy to DMARDs due to their anti-inflammatory properties and low risk of side effects.

### 1.2. Osteoarthritis

The prevalence of osteoarthritis increases with age and can be potentiated by obesity and trauma. Joint cartilage wears out with aging, and its loss causes joint pain, stiffness, and swelling [8,9,10]. All these symptoms lead to decreased physical function and deformity of the joint [11]. Despite many studies, the pathophysiology of OA is still poorly understood. Nevertheless, inflammation is the main pathophysiological process causing disease progression. Mediators of the inflammatory process are produced by osteoblasts, synoviocytes, and macrophages [12,13,14].

Frequently, chondrocytes become a target of cytokines, stimulating their apoptosis, and excessive cartilage erosion is observed [15,16,17,18]. In this process participate also extracellular matrix metalloproteinases (MMPs) and aggrecanases, which are responsible for the degradation of aggrecan and three types of collagen [19,20,21,22].

In a healthy joint, there is a balance between anabolic and catabolic cytokines, which determines a stable cartilage metabolism [23]. Unfortunately, in OA, a predominance of pro-inflammatory (catabolic) cytokines and tumor necrosis factors over regulatory cytokines and anabolic growth factors is observed [24].

The inflamed joint also exhibits overexpression of cyclooxygenase-2 (COX-2), nitric oxide (NO), and reactive oxygen species (ROS), which emphasize the role of oxidative stress in the development of OA [25,26,27]. The significance of signaling pathways like ERK-, JNK-, p38-MAPK, NF-κB, and Wnt, which control the production of pro-inflammatory factors, should also be underlined [28,29,30,31]. Thus, the complicated network participating in OA development is linked to inflammation, showing a myriad of intriguing pathologic processes (Figure 1).

### 1.3. Rheumatoid Arthritis

RA is a chronic systemic immune-mediated disease that primarily causes synovial tissue inflammation in hands and feet joints. Untreated RA causes disability and early mortality [32]. In RA, immune system modulation is compromised, leading to an overactive inflammatory response and aberrant inflammation control mechanisms. As a result, joint antigen presented by antigen-presenting cells (APCs) is recognized by CD4^+^ lymphocytes, which are activated and begin to produce pro-inflammatory cytokines contributing to the damage of bone and cartilage [33,34,35]. These cytokines (e.g., TNF) directly or indirectly activate synoviocytes and chondrocytes, which release metalloproteinases which break down the extracellular matrix (ECM) [35,36]. Additionally, CD4^+^CD28-T cells have autoreactive and cytotoxic properties and can, thus, become capable of destroying articular and periarticular tissues [37]. Another pathophysiological mechanism present in RA development is oxidative stress, which stimulates the inflammatory process. Cell balance is disturbed by increased amounts of free radicals and an inflow of neutrophil granulocytes, leading to joint damage [38,39]. Oxidative processes are accompanied by the stimulation of pro-inflammatory cytokine-activated signaling pathways like MAPK, NFκB, PI3K/AKT, JAK/STAT, and Wnt. The activation of these pathways results in the proliferation of synoviocytes and the stimulation of innate and adaptive immune system cells and mesenchymal cells. As a result, TNF-α, IL-1, and IL-6 are activated, causing synovial inflammation and enhanced angiogenesis [40].

### 1.4. Ginger Phytochemistry and Rationale for Further Review and Investigation

For over 3000 years, people have used ginger (*Zingiber officinale Roscoe*), a member of the Zingiberaceae family, as a spice and herbal remedy. Its rhizome is a particularly valuable source of bioactive substances used in Ayurvedic and Chinese medicine [41,42]. The pharmacological properties of ginger are linked to numerous active phytocompounds belonging to phenolics and terpenes. The rhizomes of ginger plants include two different types of compounds. The first is non-volatile oleoresin, a source of ginger’s pungent taste, and the second is volatile essential oils [43]. Oleoresin includes the principal physiologically active substances of this spice, such as gingerols, shogaols, paradols, and zingerone [44]. The volatile essential oil component mainly consists of sesquiterpenes, such as α-Zingiberene, and the less-abundant monoterpenes, such as borneol, 1,8-Cineole, β-Linalool, and geranial [45] (Figure 2).

One of ginger’s components, 6-gingerol, is the main compound present in the non-processed rhizome. It has anti-inflammatory and analgesic properties [46,47]. However, it is unstable at higher temperatures, and dehydration processes lead to 6-shogaol formation [48]. According to many reports, 6-shogaol is the primary ingredient in dried ginger rhizomes and has beneficial biological properties [49,50]. The different compositions of fresh and dried ginger influence the activity of various formulas and should be considered when selecting an active compound or ginger extract for medicinal purposes. These differences in composition are affected mainly by storage, thermal processing, and extraction methods. Many studies emphasize that the quality of the achieved product depends on the drying technique. For instance, Osae et al. reported freeze-drying to be the best method of obtaining high-quality ginger essential oils [51].

Conversely, the data of Ghasemzadeh et al. demonstrated that heat-drying was the most effective technique, yielding larger quantities of shogaols, with stronger antioxidant and antibacterial properties [52]. Furthermore, the amount of phenolic compounds in an extract depends on the solvents used in the extraction process. Among these, methanol was found to have the biggest impact, while acetone had the smallest influence on phenolic compounds’ quantity, which, probably, is related to the solvent’s viscosity [53,54].

Besides the extracted product’s quality, the bioavailability of ginger components throughout the gastrointestinal system is crucial. Most currently available studies describe the oral or intravenous injection of whole-extract or isolated ginger components in animal studies. Orally administered phenolic compounds, especially those with a low polarity, undergo hydroxylation and/or glucuronide and sulfate conjugation primarily in the intestinal mucosa and microflora and, secondarily, in the liver and other tissues [55,56]. Following oral administration of 2.0 g of ginger extract, a pharmacokinetic analysis revealed a 1–3 h half-life of 6-, 8-, and 10-gingerols and 6-shogaol and its metabolites [57]. In animal studies, 6-gingerol (3 mg/kg) administered intravenously was rapidly removed, with a brief terminal half-life (7.23 min) [58]. Gingerols and their analogues have a low solubility due to their structure, which impacts their bioavailability.

The biopharmaceutical drug disposition categorization system (BDDCS) is used to classify specific components. It allows for the classification of 6-gingerol and 8-gingerol into Class II and 10-gingerol and 6-shogaol into Class IV based on their solubility in buffers with varying pH levels. Despite the low bioavailability of the mentioned compounds, they demonstrate stability in experiments using simulated gastric and intestinal fluids, which increases the possibility of using them orally [59].

Technological progress enables the enhancement of the bioavailability of ginger compounds through the use of solid lipid microparticles, liposomes, and nanoparticles [60,61,62]. Nonetheless, ginger or ginger products are typically ingested alongside one’s diet. Thus, it is critical to identify the ideal conditions for technological procedures that preserve the physiological and nutritional effects of active ginger compounds.

According to epidemiological analyses, many patients with RA consume anti-inflammatory herbs in the form of pharmaceutical extracts (pills and syrups) or as nutritional components of their diet [63]. Ginger is one of these natural remedies. It is traditionally used to treat the common cold, but various studies have also revealed positive results in inflammatory diseases or antibacterial therapy [64]. Similarly, turmeric, a member of the same family, has been used as an effective supplementation in the treatment of RA [65,66]. The polyphenols, which are secondary plant metabolites with anti-inflammatory properties, present in ginger and turmeric can inhibit pro-inflammatory pathways [67]. Additionally, they have an additive analgesic effect by influencing the transient receptor potential vanilloid 1 (TRPV1) cation channel [68]. A positive effect in reducing pain in patients with OA using herbal medicines containing compounds such as capsaicin has also been confirmed [69]. Given the numerous positive properties of ginger which could potentially alleviate the symptoms of inflammatory arthritis, we decided to review the literature on preclinical in vitro and in vivo studies as well as clinical trials using ginger extract and its active compounds.

## 2. In Vitro Preclinical Studies

In in vitro studies, fibroblast-like synoviocyte cells (FLSs) are often used as a synovial membrane model for in vitro pharmacological analysis. They maintain the equilibrium of synovial fluid and extracellular matrix in healthy joints by producing synovial and articular fluid components. However, when activated, they are responsible for synthesizing inflammatory mediators, leading to joint damage [70]. The summary of in vitro studies evaluating the effects of ginger extracts and ginger compounds on FLSs and chondrocytes is shown in Table 1.

Ribel-Madsen et al. examined the impact of betamethasone, ibuprofen, and ginger extract on the pro-inflammatory factors generated by synovial cell cultures. In their study, the cells isolated from rheumatoid arthritis, osteoarthritis, and healthy patients were activated by TNF-α. Later, a ready-made extract, EV.EXT^®^77, was used to test the anti-inflammatory properties of ginger. This extract contained active substances from *Zingiber officinale* and *Alpinia galanga*, which belong to the Zingiberaceae family (e.g., 6-gingerol, 8-gingerol, 10-gingerol, gingerdione, gingerdiol, and other aromatic compounds). Surprisingly, the extract had the same impact as betamethasone in lowering the pro-inflammatory cytokine IL-8 produced by FLSs [71]. The same extract was used in another study, in which it decreased TNF-α and COX-2 expression in human synoviocytes, together with the decreased synthesis of prostaglandin E_2_ (PGE_2_). Moreover, NF-κB and nuclear factor kappa light polypeptide gene enhancer in B-cells inhibitor (IκB-α) induction were suppressed, together with the inhibition of TNF-α and COX-2 activation [72].

Another ginger extract and its components (*Zingiber officinale* extract alone and *Alpinia galanga* extract) diminished the production of chemokines in human synoviocytes in [73]. Combining two extracts synergistically has been shown to suppress monocyte chemoattractant protein-1 (MCP-1) and interferon-gamma-inducible protein 10 kD (IP-10). In the above study, *Z. officinale* was more effective than *A. galanga*; however, the immunosuppressive effect of the two extracts was more efficient than that of either extract alone. Thus, combining various ginger components can be more effective in diminishing pain in patients with OA and RA.

One of ginger’s compounds is zingerone, which has anti-inflammatory properties. Ruangsuriya et al. proved a decrease in the p38 and JNK levels in cell culture SW1353 and explant cells from pig cartilage treated with zingerone. This observation suggests that zingerone may prevent cartilage destruction by suppressing MMP-13 expression [74]. Moreover, zingerone treatments downregulate additional genes involved in cartilage catabolism, including genes encoding cytokines such as TNF-α, IL-6, and IL-8, in a dose-dependent manner [75].

This antiarthritic effect has also been shown in 6-shogaol. This ginger component causes the apoptosis of FLSs and MH7A cells and reduces their migration, invasion, and proliferation [76]. Additionally, 6-shogaol decreases the synthesis of MMP-2, MMP-9, IL-1β, IL-6, IL-8, and TNF-α, which are overexpressed in synovial tissues. According to molecular studies, this phenolic compound suppresses the PI3K/AKT/NF-κB pathway by activating peroxisome proliferator-activated receptor gamma (PPAR-γ).

The beneficial effect of ginger also reveals the action of cedrol, a sesquiterpene, downregulating ERK/MAPK and p65 NF-κB activation in stimulated FLSs. As a result of this component, suppressed activity of pro-inflammatory mediators’ mRNA and protein expression, including TNF-α, IL-1β, COX-1, and COX-2, have been observed. This process is associated with the inhibition of PGE_2_ release. Moreover, decreased levels of MMP-13 and MCP-1 have been shown to cause osteoclastogenesis resistance [77].

Ginger’s anti-inflammatory properties have been analyzed in vitro not only on synoviocytes but also on chondrocytes. Evidence suggests that chondrocyte apoptosis, a crucial stage in OA pathogenesis, might produce ROS, accompanied by the release of pro-inflammatory cytokines and damage to the extracellular matrix [16,78]. The beneficial impacts of the ginger extract in dimethyl sulfoxide (DMSO) on oxidative stress and C28I2 human chondrocyte cell apoptosis have also been observed. Studies show that ginger extract pretreatment (5 and 25 μg/mL) has a noteworthy protective effect against ROS production and the lipid peroxidation mediated by IL-1β. Ginger derivatives also boost the mRNA expression of genes related to antioxidant enzymes, such as superoxide dismutase (SOD1), glutathione peroxidase (GPX1, GPX3, GPX4), and catalase (CAT), and lower indicators of mitochondrial death, such as the Bax/Bcl-2 ratio and caspase-3 activation [79]. An aqueous extract of ginger from a commercially available product (Aquaresin Ginger, Kalsec, Inc., Kalamazoo, MI) applied at a 10–100 µg/mL concentration has been shown to decrease PGE_2_ and NO concentrations in healthy and osteoarthritic chondrocytes [80]. Furthermore, 6-shogaol also inhibits NO production in human chondrocyte cells and, thus, can prevent chondrocyte apoptosis. Additionally, it decreases the expression of MCP-1 and IL-6 [81].

**Table 1 nutrients-16-00741-t001:** In vitro preclinical effects of ginger extracts and ginger’s active compounds on fibroblast-like synoviocytes and chondrocytes.

Type of Extract/Active Compound	Dose	Type of Cells	Results	Ref.
EV.EXT^®^77 (*Z. officinale, A. galanga*)	1 mg/mL	FLSs from healthy patients or patients with RA and OA	↓IL-8 similarly to betamethasone	[71]
EV.EXT^®^77 (*Z. officinale, A. galanga*)	100 µg/mL	FLSs from patients with OA	↓activation of TNF-α and COX-2 expression, ↓synthesis of PGE_2_ and TNF-α, and ↓induction of NF-κB and IκB-α	[72]
*Z. officinale* and *A. galanga* combined extracts	100 µg/mL	FLSs from patients with OA	↓chemokines, ↓MCP-1, and ↓IP-10; the immunosuppressive effect of combined extracts were higher than either extract alone	[73]
Zingerone	20–40 µM	SW1353 cell culture and explanted cells from pig cartilage	↓MMP-13 ↓expression of TNF-α, IL-6, and IL-8 mRNA levels	[74]
Zingerone	5–20 µM	FLSs from rats with collagen-induced arthritis	Inhibition of cell proliferation and migration; ↓ROS formation, and ↓ expression of IL-1β and IL-6 transcripts	[75]
6-shogaol	5, 10, 20, 40 µM	FLSs and MH7A cell lines from patients with RA	Apoptosis of FLSs and MH7A cells and inhibition of their migration and proliferation; ↓synthesis of MMP-2, MMP-9, IL-1β, IL-6, IL-8, and TNF-α	[76]
Cedrol	1, 5, 10 µM	FLSs from mice with collagen-induced arthritis	Downregulation of ERK/MAPK and p65 NF-κB activation, ↓MMP-13, ↓MCP-1, and ↓expression of TNF-α, IL-1β, COX-1, and COX-2	[77]
*Z. officinale* extract in DMSO	5, 25 µg/mL	C28I2 human chondrocyte cells	↓ROS, ↓lipid peroxidation, Bax/Bcl-2 ratio, and capsase-3 activity; boosted mRNA expression of genes related to SOD1, GPX1, GPX2, GPX4, and CAT	[79]
*Z. officinale* aqueous extract (Aquaresin ginger)	10–100 µg/mL	Healthy and osteoarthritic chondrocytes isolated from sow cartilage explants	↓PGE_2_ and NO (doses > 10 g/mL) in OA chondrocytes and, at low concentrations, in normal chondrocytes	[80]
6-shogaol	5 µM	Human chondrocyte cells	↓MCP-1 and IL-6 ↓ MMP-2 and MMP-9 activity	[81]

↓—indicates decreased concentration, synthesis or inhibition of processes; CAT—catalase; COX-1—cyclooxygenase-1; COX-2—cyclooxygenase-2; ERK—extracellular signal-regulated kinase; FLSs—fibroblast-like synoviocyte cells; GPX—glutathione peroxidase; IκB-alfa—nuclear factor of kappa light polypeptide gene enhancer in B-cells inhibitor, alpha; IP-10—interferon-gamma-inducible protein 10 kD; MAPK—mitogen-activated protein kinases; MCP-1—monocyte chemoattractant protein-1; MMP—matrix metalloproteinase; NF-κB—nuclear factor kappa-light-chain-enhancer of activated B cells; OA—osteoarthritis; PGE_2_—prostaglandin E_2_; RA- rheumatoid arthritis; ROS—reactive oxygen species; SOD1—superoxide dismutase type 1; and TNFα—tumor necrosis factor-alpha.

## 3. In Vivo Preclinical Studies

Since ginger components contain active phytocompounds, clinical curiosity arose around whether it effectively decreases inflammatory processes, particularly in connective tissue diseases. In animal models, an orally administered commercially available ginger extract (50 mg/kg/day) has revealed antiarthritic properties [82]. Treatment with this extract significantly decreases serum and tissue levels of pro-inflammatory cytokines such as TNF-α, IL-6, and IL-17. Additionally, the administration of ginger reduces the synthesis of COX-2 and NF-κB in articular tissues. Ginger also decreases peri-synovial inflammation, synovial hyperplasia, and cartilage destruction (Table 2).

This beneficial effect has also been revealed in aqueous ginger extract. This extract reduces the concentration of cytokines such as IL-4, IL-17, and interferon-gamma (IFN-γ) in mice serum. The primary components of this extract—zingerone, 6-gingerol, 6-shogaol, and 1,4-cineol—are responsible for this effect. At a 200 mg/kg dosage, the extract suppresses MMP-1, MMP-3, and MMP-13 expression in mouse paw tissues [83]. The same dosage (200 mg/kg) of hydroalcoholic ginger extract is more successful in lowering the majority of the measured clinical, histological, and immunological parameters of mice joint inflammation than 2 mg/kg of indomethacin [84]. If this extract is administered for a longer period of time, the levels of pro-inflammatory cytokines (IL-1β, IL-2, IL-6, and TNF-α) decrease nearly twice as much in the ginger group compared to the other group on the seventeenth day. Therefore, ginger might prevent joint tissue damage by suppressing the release of pro-inflammatory cytokines and the generation of matrix metalloproteinases. The discussed studies suggest that ginger might be a useful option for replacing non-steroidal anti-inflammatory drugs when treating inflammatory arthritis.

Interesting data published by Funk et al. compare the antiarthritic properties of two extracts: crude dichloromethane extract and a fraction which contained only gingerols and their derivatives [85]. Fractions containing essential oils and polar compounds were excluded from the study because they did not inhibit PGE_2_ production in preliminary in vitro studies. Both extracts (crude dichloromethane and gingerols) were effective in reducing inflammation in the joints after i.p. injection in rats with streptococcal cell wall (SCW)-induced arthritis. However, the dichloromethane extract effectively prevented joint degeneration and inflammation since it contained more polar chemicals and essential oils. These findings imply that both gingerols and non-gingerol compounds are bioactive and have antiarthritic properties.

The same group of scientists also decided to examine the effects of a dichloromethane extract rich in sesquiterpenoids that were free of gingerols [86]. For comparison, the adjusted dose of the extract (28 mg/kg/day) was prepared to match the amount of gingerols (25 mg/kg/day) and the crude dichloromethane extract (32 mg/kg/day of essential oils and 25 mg/kg/day of gingerols) that had been used in the previous study. The extract without gingerols had no effect on acute joint swelling (measured on the third day) but suppressed joint swelling during the chronic phase of arthritis (days 13–28). Additionally, the tested extract did not affect the number of leukocytes, neutrophils, monocytes, and lymphocytes—all of which are typically higher in animals which have received streptococcal cell wall (SCW) injections. These findings suggest that ginger’s anti-inflammatory qualities may be attributed to the synergistic actions of its volatile essential oils, gingerols, secondary metabolites, and widely researched phenolics.

Recently, it has also been noted that ginger’s components can effectively reduce arthritis in mice and rats animal models. For example, cedrol, a ginger-derived sesquiterpene, has been shown to have anti-inflammatory properties [87]. Administering cedrol intragastrically every 3 days from day 15 to day 27 ameliorated the adjuvant arthritis secondary lesion score and the paw edema volume in these models. The animals that received cedrol (20 mg/kg) had lower levels of neutrophils, TNF-a, and IL-1β. In addition, a higher dosage of 80 mg/kg diminished the proliferation of these cells in synovial tissue, inflammatory cell infiltration, and cartilage damage. This suggests that ginger’s cedrol component can inhibit inflammatory mediators of joint tissue, preventing their destruction.

Similarly to in vitro studies, 6-shogaol has revealed antiarthritic properties in vivo [76]. Intraperitoneal injection of 6-shogaol (30 mg/kg/day or 60 mg/kg/day) for 21 consecutive days has been shown to reduce joint swelling in mice with collagen-induced arthritis (CIA). Pathological staining has demonstrated that 6-shogaol reduces joint inflammation, cartilage deterioration, and bone loss. Other studies have also confirmed the beneficial anti-inflammatory effects of 6-shogaol. After oral administration of 6.2 mg/kg of 6-shogaol in 0.2 mL of peanut oil, knee edema has been shown to be reduced 28 days later. Furthermore, 6-shogaol reduces soluble vascular cell adhesion molecule-1 (VCAM-1) concentration in the blood and decreases the infiltration of leukocytes (such as lymphocytes and monocytes/macrophages) into the knee’s synovial cavity. Histopathological results have shown that the femurs of rats given 6-shogaol present well-preserved and morphologically intact cartilage lining. Thus, these findings show that 6-shogaol can inhibit the inflammatory response and prevent femoral cartilage destruction [88].

## 4. Clinical Effects of Ginger in the Treatment of Osteoarthritis and Rheumatoid Arthritis

Seven randomized clinical trials demonstrated that ginger supplementation might reduce inflammation and decrease joint pain (Table 3). Only one study indicated that ginger supplements did not impact pain relief or quality of life [89]. Two studies analyzed ginger’s effect on RA, while others analyzed it on knee osteoarthritis. In all trials, the intervention procedures lasted between two weeks and three months (both stay and control-placebo groups). Two studies showed that consuming 750 mg of ginger powder twice daily could suppress the expression of several genes linked to inflammation and immune system activation in patients with active rheumatoid arthritis. At the same time, these patients were receiving medications such as methotrexate, hydroxychloroquine, and prednisolone [90,91]. A similar anti-inflammatory effect was observed in OA. In this group, using 1 g of ginger powder for three months caused a decrease in the production of pro-inflammatory cytokines [92]. An anti-inflammatory effect was also observed after a three-month supplementation of ginger extract, which lowered the concentrations of NO and CRP in patients with OA. Furthermore, in the ginger-using group, NO and CRP continued to drop even after 12 months [93]. However, prospective long-term studies can elucidate the effect of ginger extracts.

One of the beneficial effects of ginger supplementation is the low risk of complications during this treatment. In every one of the above-discussed studies, ginger did not cause severe adverse effects. Only in one case heartburn was reported [89]. It was instead proven to have no acute side effects when used for three months up to two and a half years, which is significant because individuals with OA and RA frequently use pain-relieving medications for a longer period of time [94].

Ginger powder administered orally also relieves pain in patients with OA [95]. Such an analgesic effect was also observed after the topical application of 5% ginger gel, prepared from physiological aqua gel, on the knee, and this effect was maintained for three consecutive months after two weeks of application. In contrast, in the control group using physiological aqua gel, pain relief was only temporal and quickly returned to the baseline levels [96]. Thus, topical application of ginger extracts can effectively diminish knee pain in patients with OA. In established treatments of OA or RA, mild pain in one or several joints may periodically occur [97,98]. It is worth noting that, while in the course of OA local treatment may be sufficient, in the course of RA, the lack of an appropriate DMRAD therapy leads to the intensification of symptoms and may be the cause of permanent joint destruction, which can lead to patient disability [97,98,99,100]. Therefore, in RA, the use of topical drugs should not be used interchangeably with DMARD treatment. However, the advantage of the anti-inflammatory properties of ginger allows its use as an adjunctive therapy to help reduce local minor inflammation in both OA and RA, without the necessity to increase the immunosuppressant doses of DMARD in RA or NSAID in OA.

**Table 3 nutrients-16-00741-t003:** The clinical effects of *Zingiber officinale* on rheumatoid arthritis and osteoarthritis.

Author Year [Ref.]	Study Design	No. of Participants (Age)	Type of Intervention	Comparison	Result	Condition	Duration	Adverse Events
Aryaeian et al. 2019 [91]	Randomized double-blind clinical trial	70 (19–69 years old)	*Z. officinale* (750 mg) two times a day	Placebo (wheat flour)	↓ FoxP3 gene expression ↓ RORγt and T-bet gene expression	RA	12 weeks	None reported
Aryaeian et al. 2019 [90]	Randomized double-blind clinical trial	66 (19–69 years old)	*Z. officinale* (750 mg) two times a day	Placebo (roasted wheat flour)	no effect on IL2 gene expression ↓ CRP and IL-1β mRNA (ss) ↓ TNF-α mRNA (ns)	RA	12 weeks	None reported
Mozaffari-Khosravi et al. 2016 [92]	Randomized double-blind clinical trial	120 (50–70 years old)	*Z. officinale* (500 mg) two times a day	Placebo (starch)	↓ TNF-α and IL-1β after 3 months compared to the placebo group	Knee OA	3 months	None reported
Naderi et al. 2016 [93]	Randomized double-blind clinical trial	120 (50–70 years old)	*Z. officinale* (500 mg) two times a day	Placebo (starch)	↓ NO and CRP after 3 months compared to the placebo group after the end of the study, further ↓ in NA and CRP over the following 12 months	Knee OA	3 months	None reported
Kordi Yoosefinejad et al. 2021 [96]	Randomized single-blind clinical trial	40 (≥40 years old)	5% *Z. officinale* gel formulation (70% *Z. officinale* solution + physiological aqua gel) five times a week during physical therapy	Ultrasonography with physiological aqua gel	↓ pain scores after phonophoresis with ginger gel formulation than conventional ultrasound therapy; prolonged effect of phonophoresis with Z. officinale (up to 3 months) compared to the control group	Knee OA	2 weeks	None reported
Zakeri et al. 2011 [95]	Randomized double-blind clinical trial	204 (37–75 years old)	*Z. officinale* (250 mg) two times a day	Placebo (starch)	↓ pain and ↓ morning stiffness compared to the placebo group (not statistically significant)	Knee OA	6 weeks	None reported
Niempoog et al. 2012 [89]	Randomized double-blind clinical trial	60	*Z. officinale* (500 mg) two times a day	Placebo	no effect of ginger on pain, sports activity, and quality of life when compared to a placebo	Knee OA	8 weeks	Heartburn (one patient)

↓—indicates decreased gene expression, concentration or pain sensation; CRP—C-reactive protein; FoxP3—forkhead box P3; IL-1β—interleukin-1 beta; NO—nitric oxide; RORγt—RAR-related orphan receptor gamma; TNFα—tumor necrosis factor-alpha; ss—statistically significant, ns—statistically unsignificant; RA—rheumatoid arthritis, and OA—osteoarthritis.

Recent data also show that ginger has positive effects when combined with other plants with anti-inflammatory properties, such as *Curcuma longa* or *Alpinia galanga*, which belong to the same Zingiberaceae family. Table 4 provides an overview of the available trials on this topic. The data show that ginger combined with *Alpinia galanga* decreases knee pain in OA when standing [101]. The study by Bliddal et al. compares the analgesic properties of ginger extract and an herbal preparation containing turmeric extract, ginger extract, and black pepper to the effect of naproxen and ibuprofen [102,103]. The outcomes show a decrease in PGE_2_ levels comparable to administering 250 mg of naproxen four times daily. Both ibuprofen (400 mg) and ginger extract (170 mg) show a better effect than the placebo.

Recent data show that ginger, in combination with other herbs, can be an effective painkiller if applied as a skin gel. Its activity seems to be similar to topical NSAID gel in reducing pain in patients with OA [89,103]. For instance, Niempoog et al. demonstrate that using Plygersic gel (*Z. officinale* and *Z. cassumunar* extracts) reduces pain and enhances the quality of life of patients in a manner similar to 1% diclofenac [104]. An ointment containing cinnamon, ginger, mastic, and sesame oil shows comparable pain reduction to a salicylate ointment [105]. Thus, topical formulations with ginger extract or ginger with other anti-inflammatory herbs can provide a complementary therapy to conventionally used analgesic gels. The adverse effects after topical combinations are rare, and only two incidences of contact dermatitis have been observed [104,105].

The promising results of local ginger use prompt further studies to create a brand-new ginger extract formulation incorporated in a nanostructure lipid carrier (NLC). NLC is compared to diclofenac, and both molecules show a similar effect. Furthermore, a comparison of ginger in NLC and Plygersic gel (a multiherbal formula) reveals that using ginger extract in monotherapy has similar effects to complex products.

Unfortunately, data describing the concentration of active compounds in the prepared extracts and formulations are still missing and, thus, do not enable the precise comparison of their effects between one another [101,102,104,105,106]. This assessment is difficult because the analyzed extracts usually contain various amounts of active compounds depending on the extraction method. Since their composition and functionalities strictly depend on the extraction process, determining the precise technological methods of obtaining the active substance from ginger should be of utmost importance [107].

**Table 4 nutrients-16-00741-t004:** Effect of *Zingiber officinale* on osteoarthritis.

Author Year [Ref.]	Study Design	No. of Participants (Age)	Type of Intervention	Comparison	Result	Condition	Duration	Adverse Events
Altman et al. 2001 [101]	Randomized double-blind clinical trial	261 (≥18 years old)	Ginger extract EV.EXT77 from *Z. officinale* and *Alpinia galanga* (255 mg) twice a day	Placebo (coconut oil)	↓ knee pain during standing compared to the placebo group	Knee OA	6 weeks	Mild gastrointestinal effects
Bliddal et al. 2000 [102]	Randomized double-blind clinical trial	56 (24–87 years old)	Z. officinale extract, Eurovita Extract 33 (170 mg), three times a day	Placebo or Ibuprofen (400 mg)	Statistical difference between Ibupofen, Z. officinale, and the placebo, but no difference between the placebo and Z. officinale	Hip or knee OA	3 weeks	None reported
Heidari-Beni et al. 2020 [103]	Randomized double-blind clinical trial	60 (35–75 years old)	Capsule with turmeric extract, ginger, and black pepper, containing curcumin (300 mg), gingerols (7.5 mg), piperine (3.75 mg), taken twice a day	Naproxen (250 mg)	↓ PGE_2_	Knee OA	4 weeks	None reported
Zahmatkash et al. 2011 [105]	Randomized double-blind clinical trial	92 (52.2 ± 12.4 years old)	Ointment containing cinnamon, ginger, mastic, and sesame oil (2 g) three times a day	Salicylate ointment	↓ pain, morning stiffness, and limited motion after ointment use (comparable to the effect of a salicylate ointment)	OA	6 weeks	None reported
Niempoog et al. 2012 [104]	Randomized double-blind clinical trial	99	Combination of 4% Z. officinale and Z. cassumunar extract in Plygersic gel (1 g) four times a day	1% Diclofenac gel	↓ pain and ↑ QoL after application of ginger gel (an effect similar to the 1% diclofenac gel)	Knee OA	6 weeks	Contact dermatitis (one patient)
Amorndoljai et al. 2017 [106]	Randomized double-blind clinical trial	118 (50–75 years old)	Nanostructure lipid carrier loaded with ginger extract in a ratio of 5% by weight (total amount 28.8 ± 8.7 mg)	1% Diclofenac gel	↓ knee pain, ↓ stiffness, and better physical function (ss) Effect of ginger similar to diclophenac	Knee OA	12 weeks	Skin reaction (one patient)

↓—indicates decreased pain sensation and concentration; OA—osteoarthritis; QoL—quality of life; ss—statistically significant; and ns—non statistically significant.

## 5. Ginger-Loaded Delivery Systems for Arthritis Inflammatory Treatment

The incorporation of ginger extracts in drug carriers influences the final anti-inflammatory and anti-analgesic effects. An adequate drug delivery system is still under study due to the poor bioavailability of ginger extracts and their components (e.g., 6-gingerol administered enterally) and their possible gastrointestinal adverse effects, such as heartburn or reflux [108,109]. Finding such a carrier could enable the reduction in the delivered dose and, in the case of epidermal formulations, increase the permeability of ginger through the skin layers, to the site of inflammation in connective tissue. The type of encapsulation determines drug effectiveness. Encapsulating active essential substances helps preserve these compounds and boost their stability [110].

Recent data show that most analyses have focused mainly on improving the delivery of ginger extract by transdermal administration using carriers such as nanoemulsions, nanoemulgels, or hydrogels. For example, an oil-in-water nanoemulsion containing ginger extract has been applied topically in patients with RA and in an animal model of RA. Compared to free ginger extract, the nanoemulsion showed better permeability through the skin (0.206% and 2.034% after 24 h for the ginger extract and the ginger nanoemulsion, respectively) and considerable antioxidant activity [111]. In contrast, the effectiveness of another nanoemulgel with ginger extract has been analyzed, in solitary, in vitro [112]. The anti-inflammatory activity of the ginger extract (500 g/mL) was examined using a bovine albumin denaturation assay, which revealed an anti-inflammatory efficacy comparable to that of ibuprofen (73.82% and 75.68%, respectively). Moreover, an absorbency assay using a Franz diffusion cell, in which drug release ranged from 67.55–2.32% to 87.56–3.5% over a period of 5 h, allowed the estimation of drug permeability. The study in question demonstrated that ginger’s permeability through the skin is improved by the use of a drug carrier, which may lead to enhanced bioavailability. Interestingly, the possibility to co-deliver ginger oil and ashwagandha oil suspended in hydrogel has been proven; however, the penetration efficacy has not yet been analyzed for this formulation [113,114].

On the other hand, various forms of oral administration, such as nanostructured lipid carriers, have also been analyzed [115]. For this purpose, an alcoholic extract of ginger containing 38.76 ± 3.01% *w*/*w* of 6-gingerol was incorporated into NLC and administered orally to rats with Chronic Freund’s adjuvant (CFA)-induced arthritis. Compared to the group treated with the ginger extract alone, the rats that had been given the ginger formulation showed a significant reduction in their paw volume from day 5 to day 28. This reduction in inflammation and pain may have been caused by the nano size of the carrier, which enhanced the bioavailability of the ginger extract.

While nanotechnology offers an effective delivery strategy for ginger, further research on pharmacokinetics and pharmacodynamics could yield detailed insights into these formulations’ molecular pathways and bioavailability in treating inflammatory arthritis symptoms.

## 6. Conclusions

Preclinical studies have shown that the commonly used ginger and its phytochemicals, including 6-shogaol, zingerone, and cedrol, are effective antirheumatic agents. They alter signaling pathways that are important in the pathophysiology of osteoarthritis and rheumatoid arthritis and result in the suppression of pro-inflammatory cytokines. Gingerols, shogaols, paradols, and other polyphenols found in ginger, together with sesquiterpenes, have been shown to inhibit TNF-α, IL-1β, IL-2, IL-4, IL-6, and IL-17 and reduce the synthesis of MMP-1, MMP-3, and MMP-13. They also inhibit cyclooxygenase activity (COX-1, COX-2). Thus, gingerols and their derivatives can be an alternative to non-steroidal anti-inflammatory medicines, without serious gastrointestinal or renal side effects.

However, the quantities of the active compounds in the ginger products and extracts used in the clinical trials discussed in this study are not disclosed in the majority of research works. There is growing evidence that the final ratio of bioactive chemicals might vary depending on a rhizome’s geographic origin and the extraction process employed. Because of their thermal lability, gingerols quickly dehydrate to produce shogaols. Given that the bioavailability and pharmacological properties of the two molecules differ, conversion to shogaols significantly affects ginger’s bioavailability and might be a reason for the discrepancy between the studies discussed in this review. Thus, upcoming clinical trials should allow for the determining of the bioactive compounds present in ginger and the standardizing of ginger products’ components.

Taking into account the results of the clinical trials, it can be concluded that ginger in a dose of 0.5–1 g/day administered orally for 6–3 months and dermal preparations with around 5% of ginger applied for 6–12 weeks may be helpful in reducing pain, stiffness, and inflammation in patients with OA and RA (Figure 3). Nevertheless, more studies on patients with RA are needed. Moreover, the occurrence of only three cases with side effects such as heartburn and skin dermatitis allows us to conclude that ginger is essentially safe in the doses used. In addition, the lack of beneficial effects in terms of the improvement of symptoms in one out of the thirteen cases of OA studied may have been related to the quality of the extract used and its unconfirmed content of active compounds, as pointed out earlier in this review. However, studies conducted in vitro and in vivo show that not only ginger extracts but also its active compounds, including 6-shogaol, cedrol, and zingerone, play a key role in reducing inflammation. Also, the combination of *Zingiber officinale* and *Alpinia galanga* extracts should be taken into account in further studies due to the synergistic effect between these two extracts, as suggested by the results of some in vitro studies.

In addition to developing ideal conditions and technologies to ensure the physiological effect of ginger, future research should focus on improving the bioavailability of ginger compounds at the site of administration. Therefore, researchers’ attention should be turned to modern oral and dermal delivery systems to take advantage of ginger’s therapeutic potential. Many studies demonstrate that preparing epidermal formulations enhances the penetration of ginger components through the skin and increases its healing properties. Also, the preparation of ginger extract in the form of a nanostructure lipid carrier administered orally has shown a better therapeutic effect than the administration of the extract alone.

The availability and safety of ginger reveal its tremendous unexplored potential. The results of preclinical and clinical studies show ginger’s anti-inflammatory properties, meaning that it can be used as a novel additive therapy in rheumatic diseases. Future approaches should reveal what kind of ginger components are most effective in patients with inflammatory arthritis.

## Figures and Tables

**Figure 1 nutrients-16-00741-f001:**
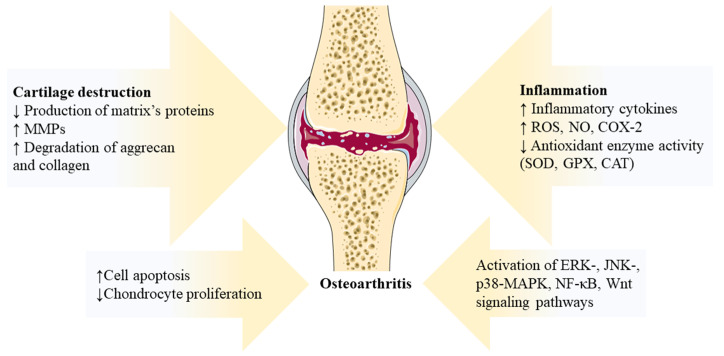
Illustration of the pathogenic processes in osteoarthritis. Arrows indicate factors affecting the inflammatory process, cartilage destruction and activation of signaling pathways occurring in osteoarthritis. CAT—catalase; COX-2—cyclooxygenase 2; ERK—extracellular signal-regulated kinase; GPX—glutathione peroxidase; JNK—c-Jun N-terminal kinase; MMPs—matrix metalloproteinases; NF-κB—nuclear factor kappa-light-chain-enhancer of activated B cells; NO—nitric oxide; p38-MAPK—p38 mitogen-activated protein kinases; ROS—reactive oxygen species; and SOD—superoxide dismutase. Resources: The figures were partly generated using Servier Medical Art, provided by Servier, licensed under a Creative Commons Attribution 3.0 unported license.

**Figure 2 nutrients-16-00741-f002:**
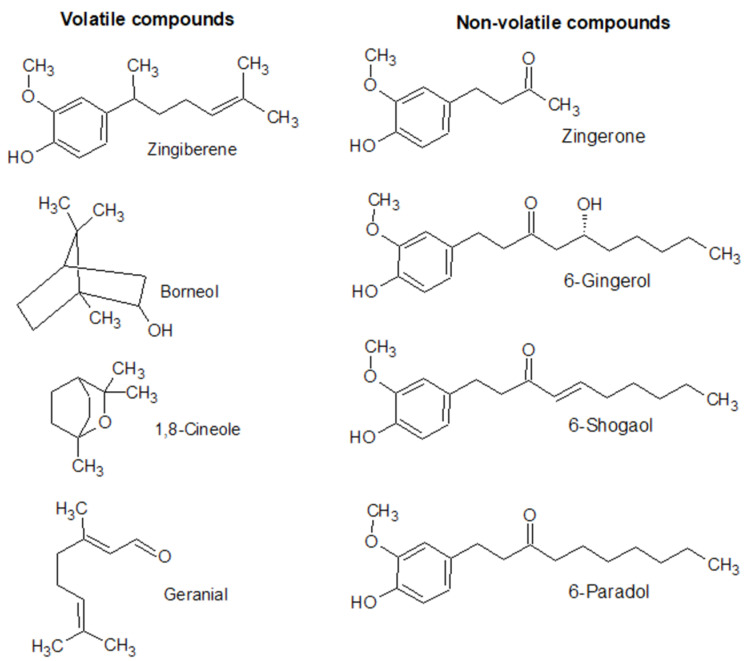
Structures of some volatile and non-volatile phytochemicals present in ginger rhizomes.

**Figure 3 nutrients-16-00741-f003:**
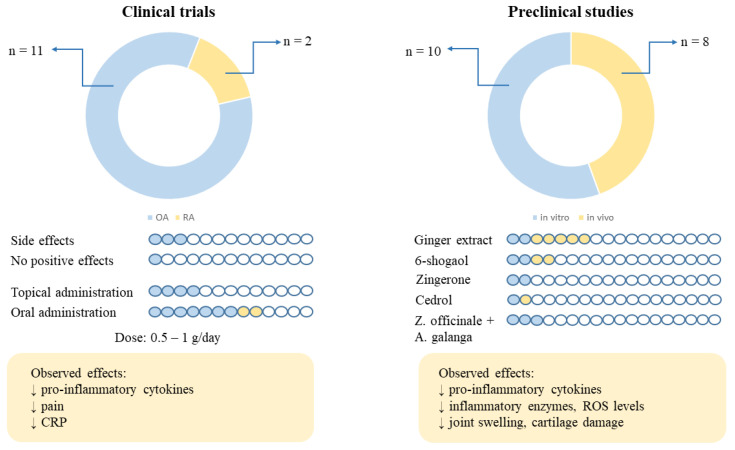
The summary of the number of clinical and preclinical studies, the number of side effects, no effects observed, route of administration, and types of ginger extracts/compounds tested. The arrows preface the number of clinical trials conducted on RA (n = 2) and OA patients (n = 11) and the number of preclinical studies conducted in vitro (n = 10) and in vivo (n = 8). CRP—c-reactive protein; OA—osteoarthritis; RA—rheumatoid arthritis; and ROS—reactive oxygen species.

**Table 2 nutrients-16-00741-t002:** The summary of the in vivo preclinical effects of ginger extracts and ginger’s active compounds.

Type of Extract/Active Compound	Dose	Route of Administration	Animal Model	Results	Ref.
Ginger extract (Gingever)	50 mg/kg	Orally for 32 days	Rats with collagen-induced arthritis	↓serum and tissue levels of TNF-α, IL-6, and IL-17; ↓synthesis of COX-2 and NF-κB in articular tissues; ↓peri-synovial inflammation, synovial hyperplasia, and cartilage destruction	[82]
Aqueous ginger extract	100, 200 mg/kg	Orally for 14 days	Mice with collagen-induced arthritis	↓concentration of IL-4, IL-17, and IFN-γ in mice serum; suppression of MMP-1, MMP-3, and MMP-13 expression in mouse paw tissues	[83]
Hydroalcoholic ginger extract	50, 100, 200 mg/kg	i.p. for 25 days (from day 7 to day 32)	Rats with collagen-induced arthritis	Doses >50 mg/kg caused suppression of IL-2, IL-6, IL-1β, and TNF-α cytokines more effectively than 2 mg/kg of indometacin; a dose of 200 mg/kg was more effective in lowering the clinical, histological, and immunological parameters of rat joint inflammation than 2 mg/kg of indomethacin	[84]
Crude dichloromethane extract and a fraction with gingerols and their derivatives	0.5–1 µL/g DMSO	i.p. (1) 4 days prior to arthritis induction and continuing until day 14 and then twice weekly; (2) from day 3 daily until day 10 and then twice weekly	Rats with streptococcal cell wall (SCW)-induced arthritis	Only the DCM extract inhibited joint destruction; the DCM extract was slightly more effective in reducing joint swelling than the gingerols-rich extract	[85]
Dichloromethane extract rich in sesquiterpenoids that are free of gingerols	1 µL/g DMSO	i.p. 4 days prior to arthritis induction and continuing until day 14 and then five days per week	Rats with streptococcal cell wall (SCW)-induced arthritis	No effect on acute joint swelling (measured on third day), but it suppressed joint swelling during the chronic phase of arthritis (days 13–28); the extract did not affect the number of leukocytes, neutrophils, monocytes, and lymphocytes	[86]
Cedrol	20, 40, 80 mg/kg	orally every 3 days from day 15 to day 27	Rats with adjuvant-induced arthritis	Amelioration in paw edema volume and arthritis score; ↓levels of neutrophils, TNF-a, and IL-1β; inhibition of T-cell and B-cell activation; a higher dosage of 80 mg/kg diminished proliferation of the above cells in synovial tissue, inflammatory cell infiltration, and cartilage damage	[87]
6-shogaol	30 mg/kg, 60 mg/kg	i.p. for 21 days	Mice with collagen-induced arthritis	Reduction in joint inflammation, cartilage deterioration, and bone loss	[76]
6-shogaol	6.2 mg/kg	Orally for 28 days	Rats with adjuvant-induced arthritis	Reduction in knee edema after 28 days; ↓concentration of VCAM-1 and infiltration of leukocytes into the knee’s synovial cavity; the femurs of rats had well-preserved and morphologically intact cartilage lining	[88]

↓—indicates decreased concentration, synthesis or inhibition of processes; COX-2—cyclooxygenase-2; DCM—dichloromethane; DMSO—dimethyl sulfoxide; IFN-γ—interferon gamma; MMP—matrix metalloproteinase; NF-κB—nuclear factor kappa-light-chain-enhancer of activated B cells; TNFα—tumor necrosis factor-alpha; and VCAM-1—vascular cell adhesion molecule 1.

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
