# Peer review of "Zingiber Officinale Roscoe: The Antiarthritic Potential of a Popular Spice—Preclinical and Clinical Evidence"

_nutrients, 2024, doi:10.3390/nu16050741_

Round 1
Reviewer 1 Report
Comments and Suggestions for Authors
this is a fine review.
I miss the practical advise for the doctors treating the patients
so add a clinical advise:
which patients can be advised to be treated (RA, OA)
when should they be treated (Beginning of the disease, later also? for which complaints?
How: orally, or local by ointment. Which ointments can be advised?
can you advise certain preparations?
Not all kinds of gingember are appropriate. please guide the doctors through the jungle of real and fake gingember treatmenst.
Should they use combination therapies.
may I suggest you to consider quoting the following papers:
Srivastava KC et al. Ginger (Zingiber Officinale) in rheumatism and musculoskeletal disorders. Medical Hypotheses 1992; 39: 342-348
Wigler I et al. The effects of Zintona EC (a ginger extract) on symptomatic gonarthritis. Osteoarthritis Cartilage 2003; 11: 783-789
Long L et al. Herbal medicines for the treatment of osteoarthritis: a systematic review. Rheumatology 2001; 40: 779-793
Leach MJ et al The clinical effectiveness of Ginger (Zingiber officinale) in adults with osteoarthriti Int J Evid Based Heathc 2008;6:311-320
Author Response
Dear Reviewer,
Thank you very much for any comments on the submitted review. As has been suggested, the conclusions section includes information on the dosage and length of use of ginger preparations, as well as their effect on the symptoms of osteoarthritis and rheumatoid arthritis, which may be a helpful tip for doctors. Also added is Figure 3 showing a summary of the number of clinical and preclinical studies conducted and described in the review. The figure also highlights the number of adverse effects that occurred, lack of effects, and the types of ginger extracts and compounds used in the studies, as well as dosage form. It also stressed that it is worth noting the beneficial effect of the combination of Zingiber officinale and Alpinia galanga. In section 4 we highlighted the use of ginger as adjunctive therapy in osteoarthritis and rheumatoid arthritis without the necessity to increase the DMARD or NSAID doses.
Also, we decided to quote the following suggested publications: Srivastava KC et al. Ginger (Zingiber Officinale) in rheumatism and musculoskeletal disorders. Medical Hypotheses 1992; 39: 342-348 and Long L et al. Herbal medicines for the treatment of osteoarthritis: a systematic review. Rheumatology 2001; 40: 779-793.
Reviewer 2 Report
Comments and Suggestions for Authors
Thank you for this review article on the antiarthritic potential of ginger in the context of osteoarthritis and rheumatoid arthritis. While the review article contains quite a bit of interesting information, it is not very accessible to the reader in its current form – see specific comments below on presentation of data overall.
16 – analyzes not analyses
Recommend to shorten and integrate section 2 on pathogenesis into the introduction section – both OA and RA have been reviewed more extensively elsewhere and some of the information is repetitive to section 1. Integrate into section 1 on introduction for why ginger is being reviewed then introduce ginger here as 4thparagraph as follows:
Paragraph 1 – inflammatory arthritides prevalence etc
Paragraph 2 – OA
Paragraph 3 – RA
Paragraph 4 – Ginger and rationale for further review and further investigation
84-86 – recommend to remove this or reword – while cartilage has historically been the focus of OA research, OA is increasingly recognized to be a multifactorial disease process in which synovial macrophages and the innate immune system play a role to perpetuate low-grade inflammation which in turn results in cartilage damage – recommend to remove and reduce this section as described above with appropriate citations
Recommend to add additional summary tables for in vitro and in vivo preclinical studies for clarity
300 – what animals? Are all preclinical studies referring to mice? This should be more clearly stated
341 – what concentration and carrier for ginger gel
357 – PGE? Not PGF?
362 – provide citation for this statement?
443 – this could be more strongly stated as it seems the major limitation to clinical application
Adding an overview image summarizing the number of studies described here would be very helpful – i.e. in vitro, preclinical, and clinical trials for OA vs RA – this information could also be included in the summary/abstract and summarized for those that found positive results vs not – overall, there is a lot of good information here but it is not very accessible to the reader in its current format
Comments on the Quality of English LanguageMinor edits requested.
Author Response
Dear Reviewer,
Thank you very much for any comments on the submitted review.
As has been suggested, lines 16, 357 has been corrected.
Also section 2 on pathogenesis of osteoarthritis and rheumatoid arthritis was shorten and integrated into the introduction section. This section is divided into the following paragraphs with revised titles: 1.1. Inflammatory arthritides prelevance and treatment, 1.2. Osteoarthritis, 1.3. Rheumatoid Arthritis, 1.4. Ginger phytochemistry and rationale for further review and further investigation. Paragraph 1.4 tells why ginger is the subject of the review. As recommended, lines 84 – 86 has been removed.
In the sections on in vitro and in vivo preclinical studies, tables (Table 1 and Table 2) have been added summarizing the studies conducted, which were discussed in the body of the review. It was explained that a study was conducted on mice and rats and which ginger gel was used in the study ( line 300 and 341 respectively). On line 362 a citation has been provided.
The conclusions section, as suggested, includes information on the dosage and length of use of ginger preparations, as well as their effect on the symptoms of osteoarthritis and rheumatoid arthritis, which may be a helpful tip for doctors. Also added is Figure 3 showing a summary of the number of clinical and preclinical studies conducted and described in the review. The figure also highlights the number of adverse effects that occurred, lack of effects, and the types of ginger extracts and compounds used in the studies. This paragraph also highlights a limitation in the use of ginger, namely its poor bioavailability. Among other things, improved oral administration was mentioned.
Round 2
Reviewer 2 Report
Comments and Suggestions for Authors
Thank you to the authors for responding to the reviewers' comments. The manuscript appears improved in its current form.